# Genomic and Phenotypic Variations Among Thai-53 and *Mycobacterium leprae* Clinical Isolates: Implications for Leprosy Pathogenesis and Research

**DOI:** 10.3390/pathogens13110986

**Published:** 2024-11-12

**Authors:** Tiago Araujo Gomes, Tatiana Pereira da Silva, Edson Machado, Sidra Ezidio Gonçalves Vasconcelos, Bruno Siqueira Mietto, Daniela Ferreira de Faria Bertoluci, Patricia Sammarco Rosa, Roberta Olmo Pinheiro, Philip Noel Suffys, Letícia Miranda Santos Lery, Flavio Alves Lara

**Affiliations:** 1Laboratório de Microbiologia Celular, Instituto Oswaldo Cruz, Fundação Oswaldo Cruz, Rio de Janeiro 21040-360, Brazil; tgomes1989@gmail.com (T.A.G.); leticia.lery@ioc.fiocruz.br (L.M.S.L.); 2Rede de Micobactérias, Programa Inova-IOC, Instituto Oswaldo Cruz, Fundação Oswaldo Cruz, Rio de Janeiro 21040-360, Brazil; tpsilva2000@gmail.com (T.P.d.S.); edson.filho@fiocruz.br (E.M.); vasconcellossidra@gmail.com (S.E.G.V.); rolmo@ioc.fiocruz.br (R.O.P.); psuffys@gmail.com (P.N.S.); 3Laboratório de Hanseníase, Instituto Oswaldo Cruz, Fundação Oswaldo Cruz, Rio de Janeiro 21040-360, Brazil; 4Laboratório de Biologia Molecular Aplicada a Micobactérias, Instituto Oswaldo Cruz, Fundação Oswaldo Cruz, Rio de Janeiro 21040-360, Brazil; 5Laboratório de Neurobiologia, Instituto de Ciências Biológicas, Universidade de Juiz de Fora, Juiz de Fora 36036-900, Brazil; bruno.mietto@ufjf.edu.br; 6Divisão de Pesquisa e Ensino, Instituto Lauro de Souza Lima, Bauru 17034-971, Brazil; danibertoluci@hotmail.com (D.F.d.F.B.); prosa@ilsl.br (P.S.R.)

**Keywords:** leprosy, non-synonymous mutations, single nucleotide polymorphs, SNPs, Thai-53, Br014-03, Br014-01

## Abstract

Throughout *Mycobacterium leprae’s* (*M. leprae*) evolutionary trajectory, nearly half of its genome was converted into pseudogenes. Despite this drastic reduction in genetic content, the genome sequence identity among *M. leprae* isolates worldwide is remarkably high compared to other pathogens. In this study, we investigated the genotype and morphotype of three *M. leprae* strains: the reference strain Thai-53 (genotype 1A), and two clinical isolates from Brazilian leprosy relapse patients, which were Br014-03 (genotypes 3I) and Br014-01(4N). We compared their genome sequences and their interaction with human Schwann cells from the ST88-14 lineage and with human primary macrophages. On the genetic level, we observed over a hundred missense mutations in the three strains, translated into significant phenotypic changes such as: prolonged doubling time, altered cytokine induction, reduced interaction rates, and decreased intracellular viability in Schwann cells. Our findings underscore the concept that despite their 99.992% identity, even small genomic disparities in *M. leprae* genomes can elicit substantial alterations in bacilli interaction with host cells and subsequent immune responses. Consequently, our data could lead to better comprehension of correlation between pathogen mutations and the diverse clinical manifestations observed in leprosy patients.

## 1. Introduction

Leprosy, caused by *Mycobacterium leprae* or *Mycobacterium lepromatosis*, is a chronic infectious disease affecting multiple systems, prominently the skin and peripheral nervous system, while also impacting the mucosa of the respiratory tract and eyes [1]. Its clinical spectrum ranges from the paucibacillary form, characterized by limited bacillary presence and often isolated skin lesions, to the multibacillary form, marked by extensive bacillary load and multiple skin lesions harboring foamy macrophages [2]. The molecular bases responsible for this range of clinical manifestations are not clear, but there seems to be a consensus that this contrasting polarization primarily resides within the impact of host’s genetic and epigenetic repertoire on cellular immune activation and cytokine production, given the minimal genetic variability of *M. leprae* strains [3].

Paucibacillary leprosy is typified by the presence of an antigen-specific cellular immune response, featuring pro-inflammatory cytokine production such as IL-2, IL-6, IL-17, TNF, and IFN-γ, leading to well-organized granulomas and containment of the infection mediated by macrophages and CD4+ T lymphocytes [2,3]. Conversely, the multibacillary form elicits a Th2 response characterized by increased secretion of IL-4 and IL-10, decreased antigen-specific cellular immune response, minimal granuloma formation, and dominance of CD8+ T lymphocytes, despite the presence of bacilli-laden foamy macrophages within lesions [2,3]. 

Comparative genomic sequence analysis of seven *M. leprae* strains from different geographic locations demonstrated a remarkably conserved genome, presenting a very low rate of single nucleotide polymorphisms (SNPs), approximately one in every 28 kb [4]. Other pathogens such as *M. tuberculosis*, *Salmonella typhi*, and *Helicobacter pylori*, present much higher SNPs frequency in their genomes, with one in every 3 kb, 1 kb, and 0.003 kb, respectively [5,6,7]. 

Comparing 175 clinical and laboratory samples from 21 countries of the five continents, Monot and collaborators observed sixteen SNP types: genotypes 1A to 1D being predominant in Central Asia, genotypes 2E to 2H, being the rarest and found in the regions of Malawi, Ethiopia, north from India/Nepal, and New Caledonia, genotypes 3I to 3M predominant in North Africa, Europe, and the Americas, and genotypes 4N to 4P mostly located in West Africa and the Caribbean regions [8].

Despite the presence of these four main genotypes, when comparing strains from Brazil, Thailand, India, and the United States, very little genomic diversity was observed; they were 99.995% identical, which suggests that leprosy arose from a single clone and that it recently went through an evolutionary bottleneck [9]. 

For this reason, over the last few decades, researchers around the world have used the Thai-53 strain with genotype 1A as a model in *M. leprae* studies. Isolated in 1981 at the Leprosy Research Center in Japan, this strain was distributed to other centers such as the National Hansen’s Disease Program in Louisiana and the Lauro de Souza Lima Institute in Brazil [10].

From 2017 to 2021, 119,698 new cases of leprosy were diagnosed in Brazil, the majority represented by subtype 3I and subtype 4N, which arrived through European colonization and the West African slave trade, respectively [4,8,11,12]. For this reason, in the present study, we compared the interaction of the Thai-53 strain and two Brazilian clinical strains: Br014-03 and Br014-01, belonging to the genotypes 3I and 4N, respectively [13], with human Schwann cells and macrophage cultures. We compared their capacity to infect and survive, and to modulate cytokine production in human Schwann cells and macrophages lineages, covering the main features of *M. leprae* infection [3]. Our objective was to determine if genetic variations between these strains translate to differences in interaction on human cells.

## 2. Materials and Methods

### 2.1. Mycobacterium leprae Strains

The three strains belonging to genotypes 1A (Thai-53), 3I (Br014-03), and 4N (Br014-01) of *M. leprae* were obtained after purification from the footpads of infected nude mice, after approximately six months of infection. Purification was carried out according to the protocol already described [14]. Thai-53 was donated by Dr. J. L. Krahenbuhl from Hansen’s Disease Program at LSU-SVM to Instituto Lauro de Souza Lima (ILSL) two decades ago and maintained in nude mice footpads according to the LSL Animal Use Ethics Committee (reference number 001/20).

The Br014-01 strain was isolated from a Brazilian male leprosy patient in 2008, resident in the northern region and suffering from disease relapse since 1980, when monotherapy was started with dapsone (DDS). His second diagnosis was in 1990 when a multidrug therapy was initiated that was composed of 24 doses of dapsone, rifampicin, and clofazimine (WHO-MDT). The patient was diagnosed with leprosy for a third time in 2008 and had been treated at that time with 12 doses of WHO-MDT. Histology of skin lesions in 2008 confirmed lepromatous leprosy with a high bacillary load. Inoculation into mouse footpads presented growth of bacilli in the presence of dapsone and rifampicin and genome sequencing confirmed their drug resistance profile to both drugs [13]. During treatment, the patient developed neuritis and presented erythematous plaques, arthritis, and nodular reaction (associated type 1 and 2 reactions). In 2020, the strain was tested again in mice fed with dapsone, rifampicin, ofloxacin, and clofazimine and confirmed resistance against dapsone and rifampicin but was sensitive to ofloxacin and clofazimine.

The Br014-03 strain was isolated from a Brazilian male patient from the southeast region of the country in 2013 who had been suffering from leprosy relapse since 1953. Clinical examination showed total madarosis, gynecomastia, painless thickened peripheral nerves, claw hands, symmetrical edema, and diffuse erythematous nodules. Histopathological examination of skin lesions confirmed active multibacillary leprosy with high mycobacterial load. Inoculation into mouse footpads demonstrated resistance to dapsone and rifampicin while genome sequencing showed an *in silico* resistance profile to dapsone, rifampicin, and ofloxacin [13]. At that moment, the patient was treated with 300 mg monthly and 50 mg daily doses of clofazimine, a 500 mg daily dose of clarithromycin, and a 100 mg daily dose of minocycline for 24 months. The patient progressed well, with improvement of the papular lesions and without new complaints or lesions. In 2016, the patient was discharged from treatment with suspension of medication. In 2020, Br014-03 was tested again in mouse footpads for dapsone, rifampicin, ofloxacin, and clofazimine and was found resistant to dapsone, rifampicin, and ofloxacin but susceptible to clofazimine.

### 2.2. Mycobacterium leprae Quality Control

After purification of the *M. leprae* strains from the mouse footpad, the presence of contaminating microorganisms was verified in LB medium and blood agar plates. The number of *M. leprae* was quantified after staining using the Ziehl–Neelsen technique and viability was measured using the Live/Dead^®^ kit (Thermo Fisher Scientific, Waltham, MA, USA) following the manufacturer’s instructions. All *M. leprae* preparations were adjusted to 10^9^ bacilli/mL and batches with viability below 80% were discarded. The doubling time (G) was calculated as described by Levy [15]. Briefly, G was calculated as the number of days between inoculation and nude mice footpad harvest divided by the number of doublings between inoculation and harvest. The number of doublings was calculated as the base-2 logarithm of the fold-increase in the number of bacteria.

### 2.3. Mycobacterium leprae Viability Analysis via Quantitative Real-Time PCR

After 24 h of infection, cells were washed three times and intracellular or associated *M. leprae* viability analysis was performed as described elsewhere [16]. Briefly, this method estimates the number of live bacilli by quantifying the rate 16S rRNA cDNA/DNA via qPCR using a Taqman^®^ MGB probe in the StepOnePlus^®^ machine (Applied Biosystems, Waltham, MA, USA). Although all cells were infected with the same number of bacteria, which viability was certified as 80% or above via Live/Dead staining, the strains presented different rates of association with Schwann cells. The 16S rRNA cDNA/DNA ratio is an effective way to adjust viability (cDNA) by genomes quantity (DNA), normalizing possible experimental discrepancies between strains.

Since *M. leprae* does not grow in medium, the following method is a sensible way to detect *M. leprae* viability inside cell cultures and clinical samples and has been used commonly during the last decade. Briefly, after 24 h of infection of the cell cultures with the three *M. leprae* genotypes, DNA and RNA were extracted using 500 µL of TRIzol^®^ (Invitrogen, Waltham, MA, USA). RNA extraction was carried out with 100 µL of chloroform:isoamyl alcohol (24:1). After centrifugation at 12,000× *g* at 4 °C for 15 min, the upper aqueous phase containing the RNA was collected and precipitated overnight with isopropyl alcohol (Sigma, St. Louis, MO, USA). The organic phase was stored for later DNA extraction adding 100 µL of TE buffer (200 mM Tris-HCl pH 8.0, 5 mM EDTA) and 150 µL of chloroform:isoamyl alcohol (24:1). After centrifugation at 13,800× *g* at 15 min, DNA was precipitated from the aqueous phase after adding 300 µL of isopropanol (Sigma, St. Louis, MO, USA) and incubated overnight at −20 °C. DNA contaminants in the RNA samples were removed using the TURBO DNA-free kit (Ambion, Austin, TX, USA) and complementary DNA (cDNA) were synthesized using the GoScript kit (Promega, Madison, WI, USA) according to manufacturer’s instructions.

### 2.4. Human Cell Cultures and Infection by Mycobacterium leprae

All cell lines were incubated under standard conditions (5% CO_2_, 37 °C). The ST8814 human Schwann cell lineage was isolated from a patient with neurofibromatosis type 1 and kindly provided by Dr. Jonathan A. Fletcher (Department of Pathology, Brigham and Women’s Hospital, Harvard Medical School, Boston, MA, USA). The cells were cultivated in RPMI 1640 medium (LGC biotechnology, São Paulo, Brazil) supplemented with 10% fetal bovine serum (FBS) (Cripion, São Paulo, Brazil). Infection was carried out with the three *M. leprae* strains for 24 h at the multiplicity of infection (MOI) as described in the figure legends.

Primary macrophages were obtained and differentiated from peripheral blood monocytes (PBMCs) of healthy volunteers and this procedure was authorized by the Oswaldo Cruz Foundation Ethical Committee (Approval number: 1.538.467, CAAE 55367216.0.0000.5248). For this, blood, once collected, was diluted in the same volume of sterile PBS (Thermo Fisher Scientific, Waltham, MA, USA) and 25 mL of diluted blood was then slowly transferred to a tube containing 20 mL of Ficoll Paque^®^ (Thermo Fisher Scientific, Waltham, MA, USA) and centrifuged at 2000× *g* for 30 min at 25 °C without the brake. After centrifugation, the fraction of PBMC was removed and washed twice in PBS (1800 g/10 min/4 °C with the brake). Cultures were performed with 5 × 10^6^ cells in 24 well plates, in 500 µL of RPMI medium with 10% FBS (Cripion, São Paulo, SP, Brazil) and 1% L-glutamine. After 2 h, non-adherent leukocytes were washed out with sterile PBS twice and the wells were filled with 1 mL of complete RPMI medium supplemented with 50 ng of m-CSF (PeproTech, Cranbury, NJ, USA). After six days of differentiation, cells were infected for 24 h in the MOIs as described in the figure legends.

### 2.5. Analysis of Mycobacterium leprae SNPs, INDELS and Genome Identity

SNP analysis was previously performed and published by Tió-Coma et al., 2020 [17] and indels by Benjak et al., 2018 [18]. In summary, SNPs were analyzed with VarScan v2.3.9, and indels were analyzed via Platypus v0.8.171. Each candidate SNP or indel was checked manually on an integrative genomics viewer. The SnpEff tool was used for variant annotation. This tool annotates and predicts the effects of genetic variants (such as amino acid changes). In the current analysis, we filtered SNPs and indels in Br-014-03 (3I) or Br-04-01 (4N) strains and compared to those present in Thai-53 [17]. The genomic position of the SNPs in the reference sequence of strain TN and predicted impact were plotted. Base variations in Thai-53, Br-014-03 (3I), and Br-04-01 (4N) were informed. Genes harboring SNPs were analyzed via gene ontology and the terms that were significantly enriched (*p* < 0.05) and presenting at least five hits, were selected.

Genome identity was calculated as described by Singh and Cole, 2011 [19]. Briefly, it corresponds to 100 − (100 × Y/Ref_size), where Y is the number of SNPs between compared strains, and the Ref_size was 3268203.

### 2.6. Cytokines Quantification via ELISA

Cell culture supernatants were collected at 1 day post infection and stored at −20 °C until use. Cytokine levels were determined via ELISA, using the following kits according to the manufacturer’s instructions: TNFα (ref#88-7346-88) and IL-23 (ref#88-7237-88) by Invitrogen (Thermo Fisher Scientific, Cranbury, NJ, USA) and, IL-1β (ref#900-T95) and IL-6 (ref#900-K16) by PeproTech (Thermo Fisher Scientific, Cranbury, NJ, USA). In summary, a 96-well high-binding polystyrene microplate was coated overnight at 4 °C with 50 µL/well of capture antibody and diluted in phosphate-buffered saline (PBS). Following incubation, the plate was washed three times with PBS containing 0.05% Tween-20 (PBS-T) to remove unbound antibody. To block non-specific binding, 200 µL of blocking buffer (1% bovine serum albumin in PBS) was added to each well, and the plate was incubated for 1 h at room temperature. After a further three washes with PBS-T, 50 µL of serially diluted standards and experimental supernatant, samples were added to the respective wells in duplicate. We constructed calibration curves using recombinant TNFα (4 pg/mL–500 pg/mL), IL-23 (15 pg/mL–2.0 ng/mL), IL-1β (8 pg/mL^–1^ ng/mL), and IL-6 (24 pg/mL–1.5 ng/mL). 

The plate was incubated for two hours at RT to allow antigen–antibody binding, followed by three additional washes with PBS-T. A biotinylated detection antibody (diluted as recommended by the manufacturers) was added to each well (50 µL), and the plate was incubated for 1 h at room temperature. After washing the plate five times with PBS-T, 50 µL of streptavidin-conjugated horseradish peroxidase was added to each well and incubated for 30 min at room temperature in the dark. The wells were washed another five times to remove unbound horseradish peroxidase-conjugate. For signal development, 50 µL of tetramethylbenzidine (TMB) substrate solution was added to each well and incubated for 10 to 20 min at room temperature in the dark until sufficient color development was observed. The reaction was stopped by adding 50 µL of 2M sulfuric acid to each well, resulting in a color change from blue to yellow. Absorbance was immediately measured at 450 nm using a microplate reader. Cytokine concentrations in the samples were determined by generating a standard curve based on the absorbance values of the serially diluted standards. All samples and standards were run in duplicate, and the assay was repeated in three independent experiments for reproducibility. Data were analyzed using SoftMax Pro 5.3 software (Molecular Devices, San Jose, CA, USA), and cytokine concentrations were expressed in pg/mL. In our analysis, TNFα and IL-23 assays sensitivity were 4 pg/mL and 2 pg/mL for IL-6 and IL-1β.

### 2.7. Supernatant Lactate Concentration Determination

Lactate quantification was performed in supernatants from cell cultures cultivated in RPMI medium containing 10% FBS, without phenol red. The measurements were carried out using the liquiform lactate kit (LABTEST, Minas Gerais, Brazil), following the manufacturer’s instructions.

### 2.8. Mycobacterium leprae and Host Cells Association Analysis

The three *M. leprae* strains were stained by PKH26 Red Fluorescent cell linker kit (Sigma, St. Louis, MO, USA) according to the manufacturer’s instructions and used for inoculation for association analysis performed via microscopy and cytometry. 

For microscopy analysis, cell cultures were infected with each one of the three strains at MOI 5:1 for 24 h at 33 °C in a 5% CO_2_ atmosphere. Cells were washed twice with PBS and fixed with 4% paraformaldehyde for 20 min at 4 °C. Then, cells were washed with PBS and the nuclei were stained by DAPI (Life Technologies, Carlsbad, CA, USA) for visualization under a Zeiss observer Z1 coupled to the Colibri Illumination System and AxioCam HRm camera (Zeiss, Oberkochen, Germany). The images were analyzed using Zeiss AxionVision software version 4.8.2 (Zeiss, Oberkochen, Germany), considering 10 random fields from each of the three biological replicates analyzed. 

The flow cytometry analysis was performed after 24 h of infection at MOI 5:1. Detached cells were resuspended in 400 µL of 4% paraformaldehyde and transferred to a cytometry tube with a 35 µm cell strainer lid (BD, Franklin Lakes, NJ, USA). Then, the cells were analyzed on the FACSAria™ Fusion cytometer (BD, Franklin Lakes, NJ, USA).

### 2.9. Identification via Mass Spectrometry of Differences in the Lipid Profile of Mycobacterium leprae Strains

Lipids were extracted from bacterial cells using methanol and formic acid solutions from a lyophilized sample containing 10^7^ bacilli from each of the three *M. leprae* strains and analyzed via ESI-HRMS as described elsewhere [20]. Briefly, spectra were acquired in 30 s in the mass range of 400 to 2000 *m*/*z*, and in quintuplicate for all samples. The online databases LIPID MAPS (University of California, SanDiego, CA—www.lipidmaps.org, accessed on 7 January 2023), HMDB version 3.6 (Human Metabolome database—www.hmdb.ca, accessed on 7 January 2023), METLIN (Scripps Center for Metabolomics, La Jolla, CA, USA—https://metlin.scripps.edu, accessed on 7 January 2023), and KEGG Pathways Database (Kyoto Encyclopedia of Genes and Genomes—https://www.genome.jp/kegg/, accessed on 8 January 2023) were accessed to pinpoint potential biomarkers with mass tolerance lower than 2 ppm.

Partial least squares discriminant analysis (PLS-DA) was used as the method of choice to evaluate the lipids similarities and differences between strains Br014-03, Br014-01, and Thai-53. This method uses multivariate regression techniques to extract, through the linear combination of the original variables, the characteristics that can highlight possible differences and similarities between the strains. Permutations of 2000 ions were used. The selection of characteristic lipids for each sample was carried out considering the impact that each metabolite had on the analysis through the VIP (Variable Importance in Projection) scores; this consists of the weighted average of the squares of the PLS-DA charges and considers the amount of variance observed in each dimension used in the model. As a cutoff threshold, only ions with a VIP score greater than 1.5 were analyzed.

### 2.10. Systematic Literature Review Analysis

To systematically determine which *M. leprae* strains have been employed in experimental published work involving Schwann cells or macrophages, a literature review analysis was conducted considering articles available from June 1970 to March 2024. Our PICO (patient/population, intervention, comparison, and outcomes) question was “What is the most prevalence type of *M. leprae* strain used in leprosy studies involving Schwann cells and macrophage research”. For that, the electronic database PubMed was systematically searched using two MeSH terms: “*Mycobacterium leprae* AND Strain AND Schwann cell” OR “*Mycobacterium leprae* AND Strain AND macrophage”. A total of 83 articles were retrieved after the initial MeSH screen, that resulted in 45 eligible studies that underwent a full-text assessment after applying the exclusion criteria. The exclusion criteria consisted of (i) non-experimental studies (i.e.,: review); (ii) studies not available online; (iii) studies not written in English, and (iv) duplicate publications during the search strategy. 

### 2.11. Statistical Analysis

The design of this study was not preregistered, and experiments were performed without blinding procedures. The normality of data was confirmed via the Shapiro–Wilk test in GraphPad Prism 7 (La Jolla, CA, USA). All graphs represent the mean ± standard deviation (SD) of at least three independent experiments performed in triplicate. Statistical analyses were carried out using Student’s *t*-test or one-way ANOVA with Fisher’s LSD test to compare the means of each column with the respective control. Differences were considered significant if *p* value < 0.05.

## 3. Results

### 3.1. Genomic Analysis of Mycobacterium leprae Strains

Our first effort was the analysis of genetic differences between *M. leprae* strains isolated in Brazil (Br014-03 and Br014-01) compared to those of Thai-53. Their genomes were published [13], and they were genotyped as 3I and 4N, respectively. Consequently, we designated these two Brazilian strains as Br014-03(3I) and Br014-01(4N). Their genomic identity varied from 99.9913% comparing Br014-03(3I) and Br014-01(4N), to 99.9955% comparing Thai-53 and Br014-01(4N). Even the Br014-03(3I) strain, which was previously revealed as hypermutated [13], still preserved 99.9917% genomic identity with Thai-53.

Upon more detailed comparison against the genome of Thai-53, we observed that Br014-03(3I) presented more SNPs than Br014-01(4N); the former had 271, being 134 in intergenic regions and non-coding transcripts while the latter presented 147 SNPs, being 69 in intergenic regions and non-coding transcripts (Figure 1A). Among these SNPs, 27.4% and 34% were predicted to have moderate impact (missense variants) in Br014-03(3I) and Br014-01(4N), respectively. More importantly, SNPs predicted to represent high impact (affecting start and stop codons) are rare, representing only 2.22% and 2.07% in Br014-03(3I) and Br014-01(4N), respectively (Figure 1B).

We identified SNPs modifying non-coding and intergenic regions in both Brazilian strains that are also more abundantly present in Br014-03(3I) (Figure 1B). The biological process most impacted by SNPs in both Brazilian strains were associated with cellular organization and carbohydrate derivative metabolic process. However, in Br014-03(3I), we observed more modifications in genes related to biosynthetic processes of organonitrogen compounds and nucleic acid metabolism. In Br014-01(4N), we predicted more modifications in small molecule’s metabolism and cellular biosynthetic processes, as well as cell surface structures synthesis, such as cell wall components and peptidoglycan synthesis (Figure 1C). 

Most of the genes containing high-impact SNPs related with relevant modifications in the protein polypeptide chain, were observed exclusively in Br-014-03 (3I) strain: *fadD9, dxs*, *ML1420,* and *nth*. Another two genes with high-impact SNPs were present but not exclusively in Br-014-03 (3I): *ML0472* and *ML2687*, and only the gene *ML1926* presented a high-impact SNP exclusively in Br014-01(4N) (Table 1).

These genomic variations suggest potential functional changes that could influence the lipid composition and cell wall structure of these strains. 

Nineteen indels were found upon comparing strains Br-014-03(3I) and Br-014-01(4N) to Thai-53. Of those, nine were located in intergenic regions, three in pseudogenes, and seven in coding regions. These seven indels located in coding regions were six in Br-014-01(4N) strain, and only one in Br-014-03(3I) (Table 2). 

The unique indel between Br-014-03(3I) and Thai-53 in in coding region was detected in *ML2678*, which encoded a 1000-amino acids-long conserved hypothetical protein. The indel concerns a G duplication at position 2521, introducing a frameshift that results in protein changes from Glu841. There are no experimental studies describing its possible function, but it contains a transglutaminase-like protein domain (PF08379) and a putative amidoligase domain (PF09899). 

Interestingly, Br-014-01(4N) strain presented the sequence CCGCACTGGTC instead of GCGTGATTGG in positions 443 to 452 of the *fad*D9 gene. Such change drives the amino acid substitutions at positions 148-150, but also introduced a frameshift, thus altering subsequent amino acids and possibly the activity of this Acyl-CoA synthetase protein (Appendix A). 

Br-014-01(4N) also presented a duplication of the sequence GAGGAGTTAAGTA located between positions 50 to 62 in *ctaB* gene introduced a frameshift and amino acid changes from Tyr21fs. *CtaB* is a probable cytochrome C oxidase assembly factor. Another gene altered was the *ML1334*, which encodes a possible conserved membrane protein of 273 amino acids and its role is not known. Br-014-01(4N) strain presented an in-frame deletion of the sequence between positions 322 to 411, resulting in a protein with a deletion of Ser108 to Gly137. This gene is polymorphic and has been used for molecular typing of *M. leprae* strains [21,22]. *BlaI* (*ML2063*) encodes a 142-amino acids-long possible transcriptional regulatory protein analogous of *M. tuberculosis Rv1846c,* involved in a beta-lactam-sensing signal transduction pathway, which is well conserved in mycobacteria, including *Mycobacterium leprae* [23]. The Br-014-01(4N) strain variant presents a C duplication at position 49, introducing a frameshift and protein alteration from His17fs.

Another indel detected in Br-014-01(4N) strain concerns a deletion of AAAGCCTCTAGAGACAGGC and insertion of a G at position 998 from *lipU* gene. This sequence alteration leads to the loss of stop codon and a frameshift variant. The *lipU* gene is 1008 bp long and codes for a putative lipase of 336 amino acids, located in cell wall and extracellular medium.

### 3.2. Functional Impact of SNPs and Lipidomic Variations

*Mycobacterium leprae* relies on surface lipids such as phenolic glycolipid 1 (PGL-1) for most of its interactions with and modulation of host cells. As an example, *M. leprae* attaches to Schwann cells via PGL-1 binding to laminin-2 that, in turn, binds to α-dystroglycan on the plasma membrane of the Schwann cells [24], or to rewire Schwann cell metabolism [25]. 

To explore this hypothesis, we conducted a comparative lipidomic analysis, which confirmed that the genomic differences were also reflected in the lipid profiles of the strains (Figure 1D). We observed that the three strains could be easily discriminated in a principal component analysis. Notably, Br014-03(3I) exhibited significant differences in the abundance of key lipids such as phosphatidylethanolamine (PE), phosphatidylcholine (PC), glucosylceramide (GlcCer), and/or galactosylceramide (GalCer) (Table 3).

This divergence in lipid composition aligns with the predicted protein modification of acyl-CoA synthetase *fadD9* in Br014-03(3I) and Br014-01(4N) (Appendix A). The SNPs analysis showed that the Br014-03(3I) *fadD9* presented a stop codon at position 107, translating a version almost 1000 amino acids shorter of the enzyme. Br014-01(4N) strain also presented three SNPs within the putative AMP-binding domain of *fadD9*, introducing amino acid changes at positions 148 (Ile > Thr), 149 (Val > Ala), and 150 (Gly > Ala), as well as an insertion–deletion. As a consequence, Br014-01(4N) *fadD9* possibly expresses two proteins: a 190-amino acids protein (100% amino acids identity from positions 1 to 147 to Thai-53) and a 1012-amino acids protein (100% amino acids identity from positions 177 to 1188 to Thai-53). These findings further support the idea that these SNPs could contribute to strain-specific lipid profiles.

We also observed that in the mouse footpad, both Brazilian strains presented a higher doubling time (G) when compared to the Thai-53 strain, with mean doubling times of 16.9 days for Thai-53, 88.8 days for Br014-03(3I), and 61.6 days for Br014-01(4N) (Figure 1E). These differences in growth rate among *M. leprae* strains should not necessarily have an impact on their association and viability analysis, since all bacterial inocula were adjusted to 10^9^ bacteria per mL, with viability certified as at least 80%.

These findings suggest that the genomic variations not only drive the metabolic and structural characteristics of the *M. leprae* strains but also potentially influence their pathogenicity, interaction with host cells, and growth.

### 3.3. Interaction with Schwann Cells and Macrophages

A determining process for the success of the infection and the development of neural damage, a hallmark of leprosy, is the association of the bacillus with Schwann cells, with subsequent modulation of cytokine profile and lactate release [26]. Therefore, we evaluated the interaction of PKH26-satined *M. leprae* and ST8814 human Schwann cells via fluorescence microscopy and flow cytometry (Figure 2). We observed that both Brazilian strains, when compared to Thai-53, presented a reduced capacity to associate with the ST88-14 cell line, this reduction being more prominent in the Br014-01(4N) strain (Figure 2B,D).

Moreover, the lower association of Br014-01(4N) with Schwann cell ST8814 apparently resulted in its inability to reduce cellular lactate release into the supernatant, a well characterized feature of Thai53 infection [26]. Both Brazilian strains also induced lower levels of TNF-γ than Thai-53, while Br014-03(3I) failed to inhibit the anti-inflammatory cytokine IL-23 (Figure 3B,C). In addition, Br014-03(3I) was less capable of surviving inside ST8814 Schwann cells (Figure 3D).

Another important hallmark of *M. leprae* is that it does not activate an inflammatory response upon interaction with macrophages [27]. We observed that although all *M. leprae* strains associated in a similar way with human primary macrophages (Figure 4A,B), both Brazilian strains induced more IL-1β than Thai-53, with Br014-03(3I) being prominent in inducing pro-inflammatory response, represented here by IL-6 and TNF release (Figure 4C).

## 4. Discussion

Throughout its evolutionary trajectory from a free-living mycobacterium to a mammalian cell-adapted pathogen, *M. leprae* underwent significant genomic reduction, losing nearly half of its genes [28]. This process likely occurred concomitant with a near-extinction event, resulting in a remarkable genetic homogeneity among contemporary and ancestral *M. leprae* isolates, exhibiting approximately 99.995% genomic sequence identity [8,9].

In the scientific literature, the Thai-53 and TN strains, both representatives of genotype 1A, are universally reported as models for research on *M. leprae*. However, in Brazil, the second most prevalent country for leprosy, the predominant genotypes are 3I and 4N [4,11,29]. Comparative genomic analyses of the Thai-53 (1A), Br014-03 (3I), and Br014-01(4N) strains revealed a high degree of homology, identifying only seven high-impact SNPs, and seven indels in coding regions, using Thai-53 as reference. Most of the exclusive SNPs were located in the Br014-03(3I) genome, and most of indels in the coding regions were identified in the Br014-01(4N) genome. 

Notably, the Brazilian strains exhibited polymorphisms primarily associated with cellular component organization and carbohydrate metabolism, potentially influencing their slower growth kinetics compared to Thai-53.

Moreover, it is plausible that the fastidious growth patterns observed in nude mouse footpad cultures are indicative of strain-specific adaptations to such an environment. Thai-53’s prolonged cultivation in nude mice footpad since 1981 contrasts with the comparatively shorter period of nude mice cultivation for Br014-03(3I) and Br014-01(4N) since 2013 and 2008, respectively. 

Interestingly, we observed that the Br014-03(3I) strain is even more fastidious than the Br014-01(4N) strain. This fact may be related to the Br014-03(3I) strain having been identified as hypermutated in a broad comparison study of 154 genomes from 25 countries [13]. One of these genes is the Acyl-CoA synthetase (*Fad*D9), where a stop codon was inserted in the position 107, generating a truncated short protein, with 107 instead of 1189 amino acids. Br014-01(4N) strain also presented three SNPs within the putative AMP-binding domain of *Fad*D9. These mutations could be related not only with the fastidious nature of these strains, but also explain the differences in their lipidomic profiles. 

The accumulation of more SNPs observed in this strain can be related to one mutation in the DNA excision repair gene endonuclease III (*nth*), leading to a truncated protein with 173 instead of 254 amino acids, due to a premature stop codon at position 517. In the present study, we observed that many of these SNPs were found in genes related to important bioprocesses such as biosynthesis of amino acids, nucleic acids, and nitrogenous compounds, as well as key enzymes for lipid metabolism such as fatty-acid-CoA synthetase *fad*D9, and the first enzyme of the isoprenoid biosynthesis: 1-deoxy-D-xylulose-5-phosphate synthase (DXS) [13]. 

This observation may also reflect an inherent trait of 3I and 4N strains, as indicated by Sharma et al. (2018), where a *M. leprae* clinical strain identified as 4N exhibited a greater proliferation rate compared to a 3I strain in armadillos [24], as well as the observation of higher frequency of *M. leprae* shifting from type 3 to 4 during relapse cases in Brazil’s region with high prevalence of type 3 genotype [25].

Recent investigations have elucidated *M. leprae*’s ability to infect Schwann cells, redirecting their metabolic pathways towards fatty acid synthesis, a process vital for pathogen survival [25,26]. Notably, much of this foundational knowledge stems from cellular studies utilizing the Thai-53 strain, as revealed by our systematic literature analysis. We observed that 21.5% of published studies involving Schwann cells and macrophages used the Thai-53 strain, while only one study reported the usage of other *M. leprae* DNA strains [27]. To our surprise, most studies (76.1%) did not state the type of *M. leprae* strain used, which may explain some of the discrepancies and divergences reported during leprosy infection [28].

Certainly, our understanding of these processes and its relevance to the pathology of leprosy would be very different if Br014-01(4N) or Br014-03(3I) strains had been used instead. Distinct lipidomic profiles further delineate the phenotypic divergence among the strains, potentially influencing their interactions with Schwann cells. Strain Br014-01(4N) displayed the most distinctive lipid composition and the lowest Schwann cell infectivity, possibly contributing to its reduced capacity to manipulate host cell metabolism and, consequently, lactate release.

Interestingly, the Br014-01(4N) strain presented the lowest infective capacity but the highest survival rate inside Schwann cell and hypothetically, the latter could compensate for being less infective, as the patient from which this strain was isolated presented neuronal function loss and neuritis throughout the treatment, indicating the presence of the bacillus at the nerves. In contrast, the lower capacity to survive of the Br014-03(3I) strain may once again reside in the fact that it is hypermutated, presenting changes in important anabolic pathways. 

The strain-dependent variations observed in infectivity and survival within Schwann cells suggests a nuanced relationship between *M. leprae* strains and some host cells, potentially influencing disease progression and treatment outcomes. Furthermore, despite the fact that Br014-03(3I) has been identified as a hypermutated strain, the phenotypic disparities observed also between strains Br014-01(4N) and Thai-53 underscore the importance of considering strain-specific characteristics for understanding leprosy pathogenesis.

No discernible differences were however noted in the rates of association between the strains with human primary macrophages. Intriguingly, Br014-03(3I) exhibited a more pronounced inflammatory response in these cells, indicating potential strain-specific immunomodulatory effects.

The knowledge about the inability of *M. leprae* to activate human primary macrophages was generated over the last few decades using the Thai-53 strain. For this reason, common knowledge was that *M. leprae* was not capable of inducing IL-β, IL-6 and TNF in macrophages [29]. Because of our present results obtained with the Br014-03 strain, we now propose that this does not seem to be the case for all circulating *M. leprae* strains. 

On the other hand, the limitation of this study resides in the fact that both clinical strains used here, Br014-01 and Br014-03, came from an *M. leprae* repository associated with a resistance monitoring consortium headed by the Lauro de Souza Lima Institute, a destination for skin biopsies of leprosy relapse cases from Brazil. Not coincidentally, both strains are resistant to dapsone and rifampicin, with Br014-03 also being resistant to ofloxacin. For that reason, they may not be the best representatives of their genotypes.

Our findings emphasize the necessity of expanding in vitro investigations and comparing more *M. leprae* clinical strains with different genotypes to associate genetic modifications with phenotypic variations and subsequent clinical implications in leprosy pathogenesis. Furthermore, they underscore the potential biases introduced by reliance on a single genotype strain for all leprosy scientific research.

## Figures and Tables

**Figure 1 pathogens-13-00986-f001:**
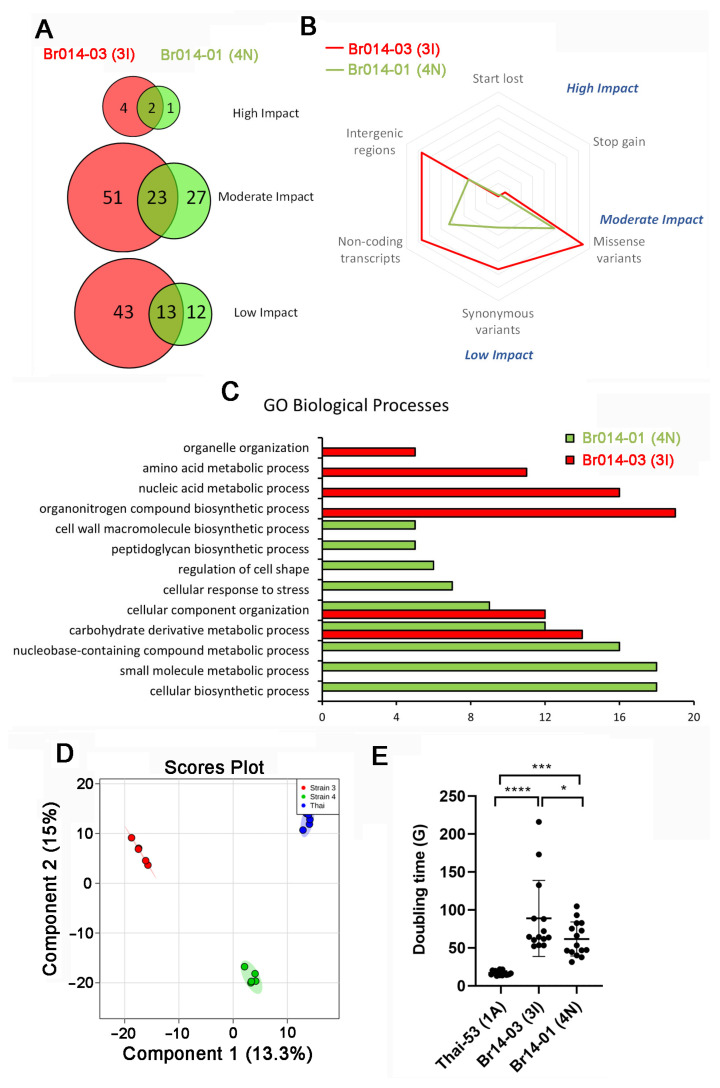
Single nucleotide polymorphisms (SNPs) between *M. leprae* strains Br014-03(3I) and/or Br014-01(4N) in comparison to Thai-53 (1A). (**A**) Predicted SNPs impact level. Overall, Br014-03(3I) presented more SNPs than Br014-01(4N), and most of the SNPs (51) were predicted to have a moderate impact. (**B**) Radar plot showing that most of the SNPs result in missense variants (moderate impact), synonymous variants (low impact), or they modified non-coding or intergenic regions. SNPs affecting start and stop codons (high impact) are rare. (**C**) Gene ontology (GO) enrichment analysis, where genes presenting SNPs in Br014-03(3I) and/or Br014-01(4N) strains were associated with GO terms. Terms significantly enriched (*p* < 0.05) and presenting at least five hits are represented. Br014-01(4N) strain SNPs are associated with cell surface structures, such as cell wall and peptidoglycan, as well as bacterial shape. Both strains present SNPs related to carbohydrate metabolism and cellular component organization. Br014-03(3I) organonitrogen compound biosynthesis is predicted to be highly impactful. (**D**) Comparative lipidomic analysis of Thai-53 (blue), Br014-03(3I) in red, and Br014-01(4N) in green, representing the lowest similarity among the strains evaluated, where each dot represents biological replicates. (**E**) The doubling time (G) of all three *M. leprae* strains calculated based in the number of days between inoculation and harvest from nude mice footpads. Statistical analysis was performed using one-way ANOVA and Fisher’s LSD multiple comparisons, where **** means *p* < 0.001, *** means *p* = 0.001 and * means *p* < 0.05.

**Figure 2 pathogens-13-00986-f002:**
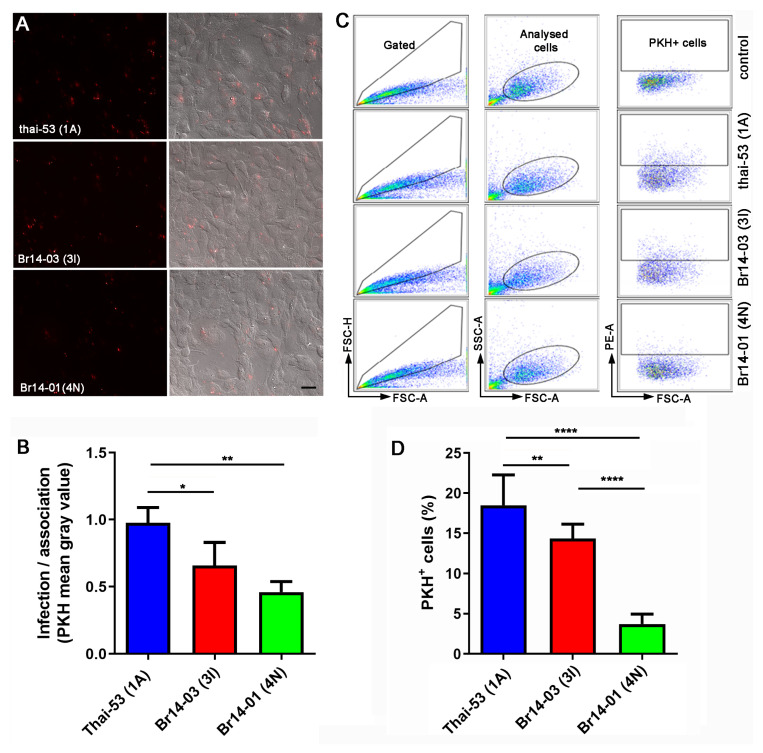
*M. leprae’s* ability to infect Schwann cells varies between strains. (**A**) Representative fluorescence microscopy of Schwann cells infected by PKH 26-stained *M. leprae* strains for 24 h with a MOI of 5:1. Scale bar means 20 µm. (**B**) Image J quantification of PKH 26 intensity in microscopy images. Values are expressed as mean ± SEM of three biological replicates. For each experiment, four technical replicates were used. (**C**) Representative FACS analysis images demonstrating the gate strategy to quantify *M. leprae*-PKH26 infected Schwann cells (PKH+ cells), demonstrating a drastic reduction in Br014-01(4N) cellular association. (**D**) Quantification of the gate PKH+ cells pointed in (**C**). Values are expressed as mean ± SEM of three biological replicates. For each experiment, five technical replicates were used. Statistical analysis was performed using one-way ANOVA and Fisher’s LSD multiple comparisons, where **** means *p* < 0.001, ** means *p* < 0.005 and * means *p* < 0.05.

**Figure 3 pathogens-13-00986-f003:**
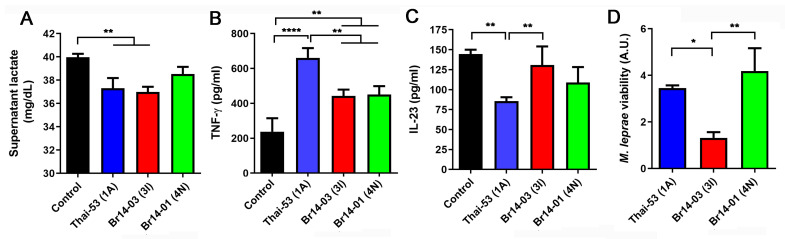
*M. leprae’s* ability to modulate Schwann cell metabolism, immune response, and survive variations between strains. (**A**) Infected Schwann cell lactate levels after infection by different *M. leprae* strains at MOI 5:1 for 24 h. (**B**) TNF-γ quantification after infection or not by different strains of *M. leprae* at MOI of 10:1 for 24 h. (**C**) IL-23 quantification after infection at MOI of 10:1 for 24 h. (**D**) Molecular determination of *M. leprae* viability via 16S rRNA levels quantification via qPCR after 48h of infection at a MOI of 5:1. Values are expressed as mean ± SEM of three biological replicates. For each experiment, at least four technical replicates were used. Statistical analysis was performed using one-way ANOVA and Fisher’s LSD multiple comparisons, where **** means *p* < 0.001, ** means *p* < 0.05 and * means *p* < 0.05.

**Figure 4 pathogens-13-00986-f004:**
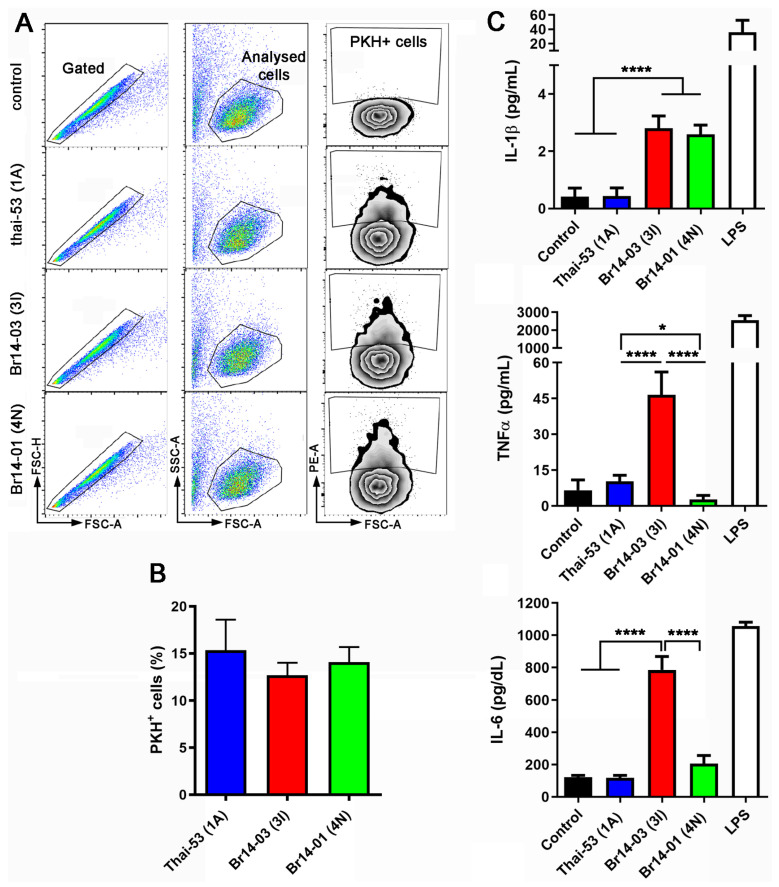
*M. leprae’s* ability to stimulate macrophages’ immune response varies between strains. Human macrophages differentiated via M-CSF were infected by *M. leprae* in a MOI 1:1 for 24 h. (**A**) Representative FACS analysis demonstrating the gate strategy to quantify *M. leprae*-PKH26 infected macrophages (PKH+ cells) (**B**) Quantification of all gates PKH+ cells, as depicted in (**A**). (**C**) Cytokines quantification in culture supernatant. Values are expressed as mean ± SEM of five biological replicates. For each experiment, at least three technical replicates were used. LPS values were excluded during the statistical analysis, which was performed using one-way ANOVA and Fisher’s LSD multiple comparisons, where **** means *p* < 0.001 and * means *p* < 0.05.

**Table 1 pathogens-13-00986-t001:** High-impact SNPs and variations between Thai-53, Br-014-03(3I) and Br-04-01(4N) strains [15].

Gene	Product	Function	SNP Impact	Position Ref Strain	Strain TN (Reference)	Thai-53 (1A)	Br-014-03 (3I)	Br-014-01 (4N)
* fadD9 *	acyl-CoA synthetase	lipid metabolic processes, long-chain fatty acid Co-A ligase activity	Stop gained	587629	G	G	A	G
* dxs *	1-deoxy-D-xylulose-5-phosphate synthase	precursor for isoprenoid, thiamin, and pyridoxol biosynthesis	Stop gained	1203135	C	C	T	C
* ML1420 *	hypothetical protein	-	Stop gained	1706535	C	C	T	C
* nth *	endonuclease III	DNA repair	Stop gained	2725968	C	C	A	C
* ML1926 *	tuberculin related peptide (AT103)	LytR: cell envelope integrity and virulence	Start lost	2312059	C	C	C	G
* ML0472 *	hypothetical protein	-	Stop gained	574185	C	T	C	C
* ML2687 *	conserved membrane protein	membrane protein, hexosyltransferase activity	Stop gained	3235177	G	A	G	G

Yellow, red and green indicates SNPs in Thai-53, Br-014-03(3I), and Br014-01(4N) strains, respectively, using *M. leprae* strain TN as reference. Data was adapted from Tió-Coma et al. 2020 [17].

**Table 2 pathogens-13-00986-t002:** Indels in coding regions of Br-014-03(3I) and Br-014-01(4N) in comparison to Thai-53.

Gene Name	Product	Impact	Position Ref Strain	Thai-53	Br14-03 (3I)	Br14-01 (4N)	Amino Acid Change
*lipU*	Possible lipase LipU	frameshift_variant&stop_lost& splice_region_variant	400762	GGCCTGTCTCTAGAGGCTTT	-	G	Gln333fs
*fadD9*	Acyl-CoA synthetase	frameshift_variant	587494	CACCAATCACGC	-	CAGACCAGTGCGG	Gly148fs
*ctaB*	Probable cytochrome C oxidase assembly factor CtaB	frameshift_variant& stop_gained	709952	C	-	CGAGGAGTTAAGTA	Tyr21fs
*ML0825c*	Probable transcriptional regulator, ArsR family	upstream_gene_variant	978674	T	-	TATTTAGGTCATCACCTTCGAGGAG	Intergenic region
*ML1334*	Possible conserved membrane protein	inframe_deletion	1587625	TCGCCTGGCCAGTACGGCTCGCCTGGCCAGTACGGCCCGCCTGGCCAGTACGGCCCGCCTGGCCAGTACGGCCCGCCTGGCCAGTACGGCC	-	T	Ser108_Gly137del
*blaI*	Possible transcriptional regulatory protein	frameshift_variant	2450703	T	-	TC	His17fs
*ML2678*	Conserved hypothetical protein	frameshift_variant	3224919	T	TG	-	Glu841fs

Red and green indicates SNPs in Br-014-03(3I) and Br014-01(4N) strains, respectively, using *M. leprae* strain Thai-53 as reference. “-” indicates no changes in comparison to Thai-53 strain.

**Table 3 pathogens-13-00986-t003:** Partial least-squares discriminant analysis (PLS-DA) comparing the three *M. leprae* strains.

Sample	Molecule	Adduct	Experimental Mass	Theoretical Mass	Error (ppm)
Br014-03(3I)	GlcCer(d41:2) and/or GalCer (d41:2)	[M + H − 2H_2_O]+	760.6475	760.6461	−1.84
	PE(O-20:0)	[M + H]+	496.3771	496.3762	−1.81
	PC(40:2)	[M + H − 2H_2_O]+	806.6426	806.6433	0.87
Br014-01(4N)	Docosanoyl-CoA	[M + H − 2H_2_O]+	1054.3782	1054.3761	−1.99
Thai53	17-phenoxy trinor PGF2α ethyl amide	[M + H]+	434.2749	434.2744	−1.15
	Phosphoenol pyruvate	[M + H − 2H_2_O]+	132.9698	132.9696	−1.50
	GlcCer(d32:2)	[M + H]+	670.5265	670.5252	−1.94

Where GlcCer means glucosylceramide, GalCer means galactosylceramide, PE means phosphatidylethanolamine, and PC means phosphatidylcholine.

## Data Availability

The original contributions presented in the study are included in the article/Appendix A, further inquiries can be directed to the corresponding author.

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
