# Peer review of "Genomic and Phenotypic Variations Among Thai-53 and Mycobacterium leprae Clinical Isolates: Implications for Leprosy Pathogenesis and Research"

_pathogens, 2024, doi:10.3390/pathogens13110986_

Round 1
Reviewer 1 Report
Comments and Suggestions for Authors
1. Line 224: "Identification by mass spectrometry of differences in the lipid profile of Mycobacterium leprae strains". Shouldn't this be a subtitle?
2. Line 269: It is recommended that the Results section also be organized under different subtitles to improve the clarity and structure of the presentation.
3. Line 274-276: How was genomic identity analyzed? Please elaborate in the Methods section on the methodology employed, including the specific software utilized.
4. Line 279: Please provide additional details in the Methods section regarding the software used for SNP analysis and the filtering criteria applied.
5. Alongside SNPs, it would be informative to know if any indels (insertions/deletions) were also analyzed.
6. Figure 1A. Clarify how the impact of SNPs is defined and explain the rationale behind not utilizing specialized software like SIFT or Polyphen2 for assessing their potential harmfulness.
7. Figure 2 & Figure 3 The inconsistency in asterisk notation for significance levels between these figures should be addressed by standardizing the convention throughout the document. Furthermore, a definition for *** is lacking and should be included.
8. It is suggested to strengthen the analysis of the correlation between the genome and lipidome in the Results section.
9. Regarding the lipidome analysis, please clarify whether it was performed on the bacterial cells themselves or on the supernatant of the bacterial culture.
10. Given the genomic and lipidomic differences analyzed among strains, it is recommended to include a discussion on the potential reasons behind these phenotypic variations.
Comments on the Quality of English LanguageOverall, the English language quality in the text is generally good.
Author Response
Comment 1. Line 224: "Identification by mass spectrometry of differences in the lipid profile of Mycobacterium leprae strains". Shouldn't this be a subtitle?
Response: The referee is correct. This sentence was numbered as subtopic 2.9 of methodology.
Comment 2. Line 269: It is recommended that the Results section also be organized under different subtitles to improve the clarity and structure of the presentation.
Response: We thank the referee for this observation. We divided the results section in 4 subsections: 3.1 Genomic Analysis of M. leprae Strains; 3.2 Functional Impact of SNPs and Lipidomic Variations; 3.3 Interaction with Schwann Cells and Macrophages.
Comment 3. Line 274-276: How was genomic identity analyzed? Please elaborate in the Methods section on the methodology employed, including the specific software utilized.
Response: Genome identity was calculated as described by Singh and Cole, 2011 (doi: 10.2217/fmb.10.153). Briefly, it corresponds to 100-(100*X/Ref_size), where X is the number of SNPs between compared strains, and Ref_size was 3268203. This information has been added to the Methods section.
Comment 4. Line 279: Please provide additional details in the Methods section regarding the software used for SNP analysis and the filtering criteria applied.
Response: SNP analysis was previously performed and published by Tió-Coma et al 2020 (https://doi.org/10.3389/fmicb.2020.01220). In summary, SNPs were analyzed with VarScan v2.3.9, and Indels were analyzed with Platypus v0.8.171. Each candidate SNP or Indel was checked manually on Integrative Genomics Viewer. SnpEff tool has been used for variant annotation. This tool annotates and predicts the effects of genetic variants (such as amino acid changes). Impact categories must be used with care, they were created only to simplify the filtering process.
In the current analysis, we have filtered SNPs in 3N or 4I strains compared to Thai. Then we filtered according to the predicted impact and plotted/listed the results. This information has been added to the Methods section.
Comment 5. Alongside SNPs, it would be informative to know if any indels (insertions/deletions) were also analyzed.
Response: We thank the referee for this observation. We access the indels of these M. leprae strains from the article published by Benjak, et al, 2018. This data is compiled now in the new Table 2, and described in the new Results section.
Comment 6. Figure 1A. Clarify how the impact of SNPs is defined and explain the rationale behind not utilizing specialized software like SIFT or Polyphen2 for assessing their potential harmfulness.
Response: We based our analysis on SNP analysis previously performed and published by Tió-Coma et al, 2020 (https://doi.org/10.3389/fmicb.2020.01220). They have used SnpEff tool for variant annotation. Such details are given for each SNP in TableS1. SnpEff details are available on GitHub (https://pcingola.github.io/SnpEff/snpeff/inputoutput/#impact-prediction). We are not aware whether SIFT and Polyphen2 could have improved performance for non-human and non-model organisms.
Comment 7. Figure 2 & Figure 3 The inconsistency in asterisk notation for significance levels between these figures should be addressed by standardizing the convention throughout the document. Furthermore, a definition for *** is lacking and should be included.
Response: We thank the referee for this correction. We correct statistical significance errors in figures legends, inserting the significance of *** in figure 1.
Comment 8. It is suggested to strengthen the analysis of the correlation between the genome and lipidome in the Results section.
Response: We rewrite this subsection to strengthen the discussion of the possible correlation between both data.
Comment 9. Regarding the lipidome analysis, please clarify whether it was performed on the bacterial cells themselves or on the supernatant of the bacterial culture.
Response: Lipids were extracted from bacterial cell extracts. We inserted this information in the methods section.
Comment 10. Given the genomic and lipidomic differences analyzed among strains, it is recommended to include a discussion on the potential reasons behind these phenotypic variations.
Response: Regarding this point, we generated a supplementary figure showing fadD9 polypeptide chain modifications in M. leprae strains, and inserted the following sentences in the results and discussion section:
line 404: "Interestingly, Br-014-01(4N) strain presented the sequence CCGCACTGGTC instead of GCGTGATTGG in positions 443 to 452 of fadD9 gene. Such change drives the aminoacid substitutions at positions 148-150, but also introduced a frameshift, thus altering subsequent aminoacids and possibly the activity of this Acyl-CoA synthetase protein (Supplementary Figure 1)."
line 450: "This divergence in lipid composition aligns with the predicted protein modification of fatty-acid-CoA synthetase fadD9 in Br014-03(3I) and Br014-01(4N) (Supplementary figure 1). This enzyme catalyzes a thioester bond between a fatty acid and coenzyme A, allowing fatty acid to be degraded for energy production or incorporated into complex lipids. The SNPs analysis show that the Br014-03(3I) fadD9 presented a stop codon at position 107, translating a version almost 1000 amino acids shorter of the enzyme. Br014-01 (4N) strain also presented three SNPs within the putative AMP-binding domain of fadD9, introducing amino acid changes at positions 148 (Ile>Thr), 149 (Val>Ala), and 150 (Gly>Ala), as well as an insertion-deletion. As consequence, Br014-01 (4N) fadD9 possibly expresses two proteins: a 190-aminoacids protein (100% aminoacids identity from positions 1 to 147 to Thai-53) and a 1012-aminoacids proteins (100% aminoacids identity from positions 177 to 1188 to Thai-53). These findings further support the idea that these SNPs and could contribute to strain-specific lipid profiles."
Reviewer 2 Report
Comments and Suggestions for Authors
Dear Authors,
I read your paper with interest. Always interested in the agent host relationship and the influence of the different factors.
I have some questions as a clinician and comments.
Introduction:
49: It is antigenic determinant responsive. It depends on how the antigen is exposed to the immune system. (Consider this also for the rest of the paper).
I love the objective of the research.
Material and Methods
You compare Brazilian resistant strains with the Thai strain. Why not with a non-resistant Brazilian strain too?
195: Why not IL10 too? To see whether resistant strains are more suppressive. Or will IL23 give the same information? You looked at IL1Beta. Which works more or less opposite IL10.
203: Did you also look at the lactate from different pathways?
As a clinician I cannot judge all the lab methods, but on my level this looks OK.
Lipid profile: Do you expect to find differences? Later I saw you found them and explained what the results where.
Results;
Good presented. For a layman in lab research other than basic immunology more or less clear.
Discussion:
408-412: interesting explanation.
425: What is the most prevalent strain in wild armadillos?
473: I see you discuss the resistance already.
Fully agree “further research in the host parasite relation is necessary particularly in the genetics”.
Author Response
Comment 1: You compare Brazilian resistant strains with the Thai strain. Why not with a non-resistant Brazilian strain too?
Response: Unfortunately, these clinical strains were clean, isolated and propagated in nude mice in an effort to identify M. leprae resistance in clinical samples. Only samples from patients with therapeutic failure are included in this effort. For that reason, all M. leprae clinical strains in culture at this moment are antibiotic resistant. There is a lack of investments in basic science regarding leprosy, for that reason we are making basic science with resources generated by patient attendance.
Comment 2: Why not IL10 too? To see whether resistant strains are more suppressive. Or will IL23 give the same information? You looked at IL1Beta. Which works more or less opposite IL10.
Response: We thanks the referee for this question. In fact, we also measured IL-10 in cultures of macrophages from the THP-1 lineage. However, contrary to what is described in the literature for primary macrophages, none of the three M. leprae strains was able to induce IL-10. Strain Br14-01(4N) actually reduced IL-10 production to below the control (Figure attached). We believe that this may be due to some deficiency in the THP-1 macrophage lineage response against M. leprae. Therefore, to avoid a discussion that would distract from the focus of the study, which aims only to analyze phenotypic differences between M. leprae strains, we decided to remove this finding from the article.
Comment 3: Did you also look at the lactate from different pathways?
Response: We determined lactate concentration only from cell suppernatant. The cells dosent acumulate lactate, due to its toxic nature. In this way, the excess of lactate generated in the regeneration of NAD+ resulting from exacerbated glycolytic activity is exported to the extracellular environment, where it accumulates. Thus, the measurement of intracellular lactate provides you with a snapshot of the metabolic state, while the measurement in the supernatant informs you of the total amount generated during the experiment time.
Comment 4: What is the most prevalent strain in wild armadillos?
Response: As far as I know, there is only one article where M. leprae isolated from armadillos was genotyped, and the totality of them belongs to the 3I genotype (10.1056/NEJMoa1010536). Similarly, M. leprae genotypes identify infecting squirrels in the UK is also 3I (10.1016/j.cub.2024.04.006).

Reviewer 3 Report
Comments and Suggestions for Authors
The manuscript entitled “Genomic and Phenotypic Variations among Thai-53 and Mycobacterium leprae clinical isolates: Implications for Leprosy Pathogenesis and Research” represents an excellent genetic investigation that aims to shed light on the impact of even small variations of Mycobacterium leprae on the host. The authors demonstrate scientific rigour and clarity of exposition in each section of the manuscript.
However, some minor additions would be useful for a better presentation of the article.
First of all, it would be desirable to include more information about the Brazilian epidemiological situation of the disease so that it can be more precisely contextualized within a broader framework involving all of South America and the rest of the world.
As far as the quantification of cytokines by ELISA kits is concerned, it is sufficient for the authors to indicate that they followed the manufacturer's instructions, but a few more explanations of the procedure would make the interpretation of the results clearer and, if it were in their possession, they could indicate the performance characteristics of the test in terms of sensitivity and specificity.
On a purely aesthetic note, figure 2 cannot be seen very well, so it would be appropriate for the authors to at least include the images also in the supplementary material in a format that can be more user-friendly.
As for the rest, the presentation of the results and the discussion, I think the authors did an excellent job and I congratulate them.
Author Response
Comment 1: It would be desirable to include more information about the Brazilian epidemiological situation of the disease so that it can be more precisely contextualized within a broader framework involving all of South America and the rest of the world.
Answer: We thank the referee for this observation. We included the following sentence at the end of introduction (page 2 line 78):"From 2017 to 2021, 119,698 new cases of leprosy were diagnosed in Brazil, majority represented by subtype 3I and subtype 4N, which arrived through European colonization and the West African slave trade, respectively."
Comment 2: As far as the quantification of cytokines by ELISA kits is concerned, it is sufficient for the authors to indicate that they followed the manufacturer's instructions, but a few more explanations of the procedure would make the interpretation of the results clearer and, if it were in their possession, they could indicate the performance characteristics of the test in terms of sensitivity and specificity.
Response: We thank the referee for this observation. We rewrite the section 2.6 of Methods, describing more detailed the ELISA procedures including sensitivity.
Comment 3: On a purely aesthetic note, figure 2 cannot be seen very well, so it would be appropriate for the authors to at least include the images also in the supplementary material in a format that can be more user-friendly.
Response: We believe that the low resolution of figure 2 is related with the upload of the preliminary document with figures pasted in Word software. The final submission will be performed uploading high resolution .tif files in the MDPI platform. At this moment we will double check the quality of figure 2 in the proof document.
Round 2
Reviewer 1 Report
Comments and Suggestions for Authors
line 95: The species name should be italicized. Please review the entire text and make revisions.
line 278: title format.
Line 218-222:The references are not presented correctly.
Please read the whole text carefully and pay attention to spelling and formatting errors.
Comments on the Quality of English LanguageMinor editing of English language required.
Author Response
Comment 1: line 95: The species name should be italicized. Please review the entire text and make revisions.
Response: The Pathogens journal uses the standard of highlighting all section subtitles in italics. In these cases, words normally highlighted in italics, such as the names of organisms or genes, should be written without italics, so that they stand out from the text in which they are inserted.
Comment 2: line 278: title format.
Response: done.
Comment 3: Line 218-222:The references are not presented correctly.
Response: We thank the referee for this correction. Although the catalog numbers are correct, the cited ELISA kits for TNFα quantification (ref#88-7346-88) and IL-23 quantification (ref#88-7237-88) were purchased from Invitrogen instead of eBioscience. The sentence were modified to:
"Cytokine levels were determined by ELISA, using the following kits according to the manufacturer´s instructions: TNFα (ref#88-7346-88) and IL-23 (ref#88-7237-88) by Invitrogen (Thermo Fisher Scientific, Cranbury, USA) and, IL-1β (ref#900-T95) and IL-6 (ref#900-K16) by PeproTech (Thermo Fisher Scientific, Cranbury, USA). "